# IMPOSE (IMProving Outcomes after Sepsis)—the effect of a multidisciplinary follow-up service on health-related quality of life in patients postsepsis syndromes—a double-blinded randomised controlled trial: protocol

Jennifer D Paratz,[1,2,3,4] Justin Kenardy,[5] Geoffrey Mitchell,[6] Tracy Comans,[7] Fiona Coyer,[8] Peter Thomas,[1,2,4] Sunil Singh,[9] Louise Luparia,[1,3] Robert J Boots[1,2]

For numbered affiliations see end of article.

**Correspondence to**
Dr Jennifer D Paratz;
j.paratz@uq.edu.au

## ABSTRACT

**Introduction:** Patients post sepsis syndromes have a poor quality of life and a high rate of recurring illness or mortality. Follow-up clinics have been instituted for patients postgeneral intensive care but evidence is sparse, and there has been no clinic specifically for survivors of sepsis. The aim of this trial is to investigate if targeted screening and appropriate intervention to these patients can result in an improved quality of life (Short Form 36 health survey (SF36V.2)), decreased mortality in the first 12 months, decreased readmission to hospital and/or decreased use of health resources.

**Methods and analysis:** 204 patients postsepsis syndromes will be randomised to one of the two groups. The intervention group will attend an outpatient clinic two monthly for 6 months and receive screening and targeted intervention. The usual care group will remain under the care of their physician. To analyse the results, a baseline comparison will be carried out between each group. Generalised estimating equations will compare the SF36 domain scores between groups and across time points. Mortality will be compared between groups using a Cox proportional hazards (time until death) analysis. Time to first readmission will be compared between groups by a survival analysis. Healthcare costs will be compared between groups using a generalised linear model. Economic (health resource) evaluation will be a within-trial incremental cost utility analysis with a societal perspective.

**Ethics and dissemination:** Ethical approval has been granted by the Royal Brisbane and Women's Hospital Human Research Ethics Committee (HREC; HREC/13/QRBW/17), The University of Queensland HREC (2013000543), Griffith University (RHS/08/14/HREC) and the Australian Government Department of Health (26/2013). The results of this study will be submitted to peer-reviewed intensive care journals and presented at national and international intensive care and/or rehabilitation conferences.

### Strengths and limitations of this study

- This is the first study to concentrate solely on patients with sepsis syndrome postintensive care discharge.
- There is a strong multidisciplinary team to provide a multitude of screening tools and interventions.
- Individualised management will be provided to patients.
- This is a single-centre study, so lacks external generalisibility.

**Trial registration number:** Australian and New Zealand Clinical Trials Registry ACTRN12613000528752.

## BACKGROUND

Greater numbers of patients are surviving in intensive care units (ICUs) but a new syndrome called the 'post-ICU syndrome' has appeared with residual physical, cognitive and psychosocial problems lasting from 5 to 15 years.[1 2] The rate of a major illness recurring and/or mortality is also increased in the year after ICU discharge, with mortality ranging from 26% to 63% at 1-year postdischarge in long-stay ICU patients (≥48 h).[3] ICU survivors additionally often have a number of problems[1–4] including debility and fatigue from loss of muscle mass, ongoing nutritional problems, difficulty in dealing with altered appearance and function, chronic pain,[5] amnesia and delusional symptoms. All of these factors have a major impact on the health and productivity of survivors and their carers, survivors' return to work rates, as well as the impact on

availability of ICU and hospital beds, surgical waiting lists, health costs and society.[6] There has been a 'call to arms' to provide improved management regimes to ICU survivors.[7] The National Institute of Health and Care Excellence (NICE) clinical guidelines[8] provided by the National Health Service (NHS) NICE recommend 'optimisation of recovery' rather than merely survival post-ICU. All of these problems are particularly present in survivors of sepsis.[9]

Sepsis is defined as a powerful inflammatory response to severe infection.[10] The annual total cost of sepsis syndromes in the USA is $16.7 billion nationally[11] with the incidence projected to increase by 1.5% per annum as patients develop more comorbidities and resistant organisms. In brief, sepsis is a major public health concern.[12]

However, the statistics above do not include the cost of ongoing disability and loss of productivity postillness. Patients with sepsis syndromes have significantly worse outcomes with lower health-related quality of life (HRQoL),[13–16] particularly in the physical domain,[17 18] chronic pain[5 19] and cognitive domain[20] compared with non-septic ICU survivors. While other critically ill/injured or disabled patients, for example, those with burns, strokes, head injuries have established follow-up regimes, patients with sepsis are not referred to formal rehabilitation and are discharged home with extremely poor function.

Follow-up ICU outpatient clinics are a recent innovation particularly within the UK.[21–23] These clinics identify and provide intervention for problems posthospital discharge including weakness, poor balance, impaired swallowing and nutrition, ongoing emotional problems and/or post-traumatic stress disorder in critical illness survivors. The one completed clinical trial on this topic[24] did not find a significant improvement in HRQoL. Researchers admitted that one discipline was not ideal and 'further work should focus on the roles of early physical rehabilitation, delirium, cognitive dysfunction and relatives'. The programme was conducted in a generic group of ICU survivors including those with only an overnight stay, and there may be greater discriminatory power in selecting only one diagnostic group with a documented poor outcome such as sepsis syndrome. Additionally some countries (Canada, Australia, South Africa) have major problems with large land area and relatively sparse population in rural areas leading to a lack of resources and problems with follow-up post-ICU.[25]

## AIMS

The primary aim of this research is to investigate whether targeted follow-up and intervention will improve HRQoL in survivors of sepsis syndromes. Secondary aims will be to investigate whether this intervention decreases readmission rates to hospital, 12-month mortality and health resource use.

This project will be an important investigation of follow-up and management of ICU patients in a subpopulation (sepsis) documented to have poor outcomes.[16] The project will incorporate a subgroup (rural patients) via telemedicine[26] in a novel method to overcome the problem of the 'tyranny of distance' in follow-up.

## Hypotheses

The hypotheses are that posthospital follow-up will improve HRQoL, decrease mortality, economic and health resource use and readmission to hospital in patients postadmission to ICU with sepsis syndrome.

## METHODS/DESIGN

This is a prospective, double-blinded, interventional, repeated measures, superiority, randomised controlled trial with concealed allocation, blinded assessors and intention-to-treat analysis. The study has been designed in accordance with the Consolidated Standards for Reporting of Trials guidelines[27] (figure 1) and the Standard Protocol items: Recommendation for Interventional Trials (SPIRIT 2013).[28] The trial will be completed in two outpatient clinics. One clinic will be in the outpatient department located at the university-affiliated tertiary hospital Royal Brisbane and Women's Hospital (RBWH), Brisbane, Australia; the other will be located at a regional hospital—the Bundaberg Base Hospital, Bundaberg, Queensland, Australia.

## Participants

Participants will be recruited from among patients being discharged from a quaternary university-affiliated ICU at RBWH, Brisbane, Australia. Participants will be randomised to one of the two groups (intervention or control) post-ICU discharge and just prior to discharge from hospital.

### Inclusion criteria

Patients included will be men or women ≥18 years of age, with a documented episode(s) of sepsis[10] (≥2 criteria of a systemic inflammatory response plus proven or strongly suspected infection, severe sepsis defined as sepsis plus organ failure, septic shock defined as severe sepsis not responding to management) and required respiratory support for longer than 48 h. Enrolment in the study should be within 1 month of discharge from hospital.

### Exclusion criteria

Patients with neurological injuries, spinal injuries and burns will be excluded, as these patients have existing rehabilitation programmes and community support groups. Patients with haematological conditions or requiring palliative care post-ICU will also be excluded. Patients with psychiatric and/or mental disabilities that preclude them from understanding the questionnaires and non-English speaking patients will also be excluded.

**Figure 1** CONSORT flow diagram.

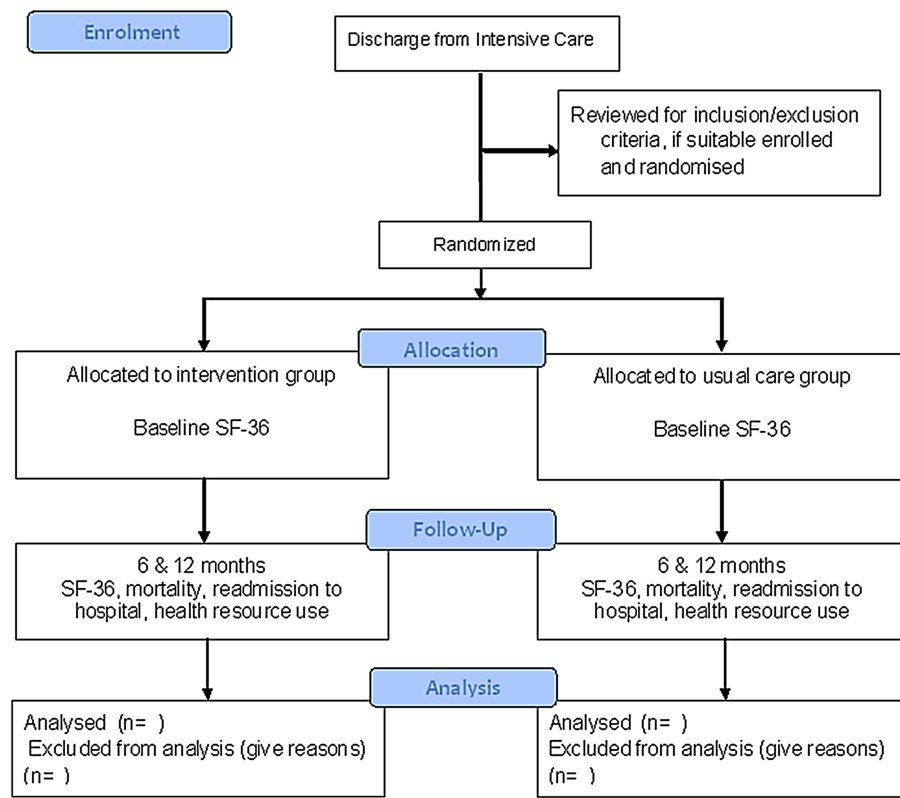

## Randomisation

Participants will be randomised by opaque sealed envelopes after enrolment. Numbers on the envelopes will be generated by a computer-generated randomisation (http://www.randomization.com) table based on blocks of four to assign patients to either the intervention or the usual care group. The randomisation sequence will be concealed from consent designee staff and protected by an electronic password. Participants will be stratified based on a score >3 on the Cumulative Illness Rating Scale (CIRS-G).[29]

We are stratifying patients as most patients who develop sepsis syndromes have a number of comorbidities, but there can be a small number of relatively young, healthy patients who are affected. The allocation sequence will be generated by the research assistant attached to the trial. Patients will be consented by the chief investigator (JDP) and assigned to interventions by a second investigator (PT).

## INTERVENTION
### Intervention group

Patients in the intervention group will attend a follow-up clinic two monthly for up to 6 months after discharge from the hospital. Screening instruments will be utilised on the first visit and appropriate management and referral provided. These will involve: overall medical review involving all systems assessment, medications, vital signs, physical activity assessment, muscle strength,[30] assessment of mobility (*modified Rivermead Mobility Index*),[31] balance assessment (*Berg Balance measure*),[32] discussion with carer

as to concerns (*Zarit Burden Interview*),[33] screening for chronic pain syndromes (*Brief Pain Inventory*),[34] nutritional review, screening with the (*PTSD*) *Checklist-Civilian Version* (for post-traumatic stress syndrome),[35] *generalised anxiety disorder (GAD-7)*[36] and *The Patient Health Questionnaire-9 for depression.*[37] Following the results of the screening and team discussion, participants and/or carer will be referred to appropriate agencies including the general practitioner, specialist, physiotherapist, nutritionist, speech pathologist, exercise physiologist, psychologist and/or occupational therapist. These instruments are screens for post-ICU problems that have been identified in previous studies.[13–20] Members of the research team include an intensivist/thoracic physician, two specialist respiratory physiotherapists, a psychologist, a senior nurse researcher, a professor of community and general practice and a health economist/community physiotherapist. This individualised management is in line with recommendations of the NICE clinical guidelines.[8]

## Standard care

Patients in the control group will have usual care, that is, they will be referred to specialist outpatient clinics and/ or general practitioners on discharge.

## Subgroup—rural patients

A small number of patients (10) originally managed in the major quaternary level ICU will be discharged to a rural area. We will test the feasibility of providing follow-up care to these patients. These patients will also be randomised to either the intervention or the usual

care group. The patients will be interviewed via telemedicine (videoconferencing)[26] and a research assistant will be appointed to complete other screening tools and refer them to appropriate agencies. If specialist services are required, for example, chronic pain clinic, these may also be accessed via telemedicine.

## OUTCOME MEASURES
### Primary measure
The primary measure will be HRQoL as measured by the Short Form 36 health survey V.2 (SF36V.2)[38] which demonstrates content and construct validity, sensitivity and responsiveness to change in many patient groups[39] including ICU and postsepsis survivors.[40] Pilot testing of the SF36 in this population by this team has been conducted in a current trial of inpatients postsepsis and has shown excellent completion rates at 6 months by telephone.[41]

### Secondary measures
Patients' readmission rates to hospital (medical record data), 12-month mortality (only expectedtrend) and economics and healthcare resource use. The latter will be monitored through examination of patient medical records, patient self-report and through Medicare/Pharmaceutical Benefits Scheme data extraction. The SF6D[42] is a preference based six-dimensional health state measure suitable for economic evaluation that is derived from the SF36. The SF6D defines 249 health states, and a summary utility score for the SF6D will be calculated using UK weights.

Individual patient productivity will be measured using a work/activity impairment questionnaire, the Work Productivity and Activity Impairment Questionnaire (WPAI).[43]

The outcome measure of SF36V.2 will be taken at baseline, 6 and 12 months postenrolment in the study. All other measures (mortality, hospital readmission and economic and health resource use) will be taken at 6 and 12 months postenrolment.

### Sample size
Power calculations are based on SF36 using previously published Minimally Clinically Important Differences (MCIDs) for the SF36 physical domain scores of 7.8 points (SD 15) in our pilot trial on rehabilitation in sepsis.[41] The sample size, based on a two sample comparison of means was calculated at 78/group for the SF36 at 90% power with a two-sided, α error level of 0.05. Allowing for an almost 30% loss to follow-up (mortality, dropouts), a total of 204 participants will be required.[44]

### Planned statistical analysis
#### Primary outcomes
A baseline comparison of demographics, length of stay in ICU, severity of illness, Acute Physiological and Chronic Health Evaluation score (APACHE II) on admission to ICU[45] and baseline measures will be carried out between each group, using a combination of t tests and $\chi^2$ test.

Generalised estimating equations (GEE) will be used to compare the SF36 domain scores between groups and across time points. The GEE is a flexible technique that takes into account the correlations among individuals in longitudinal study designs and can account for missing data without a need for imputation techniques.[46]

### Secondary outcomes
Mortality will be compared between groups using a Cox proportional hazards (time until death) analysis.[47] Time to first readmission will be compared between groups by a survival analysis. Utility estimates derived from the SF6D will be analysed using a GEE. Healthcare costs will be compared between groups using a generalised linear model to account for the non-normal distribution. Economic (health resource) evaluation will be a within-trial incremental cost utility analysis with a societal perspective.[48] Direct costs included will be: cost of care provided in hospitals, by general practitioners, home nursing, complementary (allied) health providers, alternative health providers, opportunity cost of unpaid carers, cost of transportation to programme and to other health services and cost of pharmaceuticals. Indirect costs include loss of productivity due to disease, such as time lost from work or providing informal care to another individual. All costs will be valued using market costs where available and productivity costs will be valued using actual patient wage rates.

The subgroup of rural patients will be analysed in the full analysis and subgroup results will be compared with metropolitan patients.

### Data management
A custom-designed database will store de-identified patient data on a secure password protected file. This will be entered from hard copies of the scoring sheets. Data will be reviewed by research office staff twice yearly and cross referenced with stored hard copies. A data management committee comprising a senior ICU specialist, a senior critical care nurse and a senior physiotherapist will review the data every 6 months.

### Contingencies
If participant recruitment does not reach the required sample size in 2 years, the study can be extended for a further 1–2 years.

### Methods for protecting against other sources of bias
This study will be double blinded with the assessors, data analysts and a number of investigators (TC, FC and GM) blinded from group allocation. There will be concealed allocation of participants. Participants will be stratified for CIRS-G.[29] Participants when being consented will be told that they will be in one of the two groups which will

be followed up posthospital and they may need to attend an outpatient clinic. Patients are unaware of the usual follow-up posthospital so should not feel they are receiving an increased/decreased level of care. Both groups will be contacted by telephone for completion of HRQoL life instruments at the required stages. Participants may have varying severity of illness, lengths of stay and/or time on mechanical ventilation in ICU; however, this will be compared between groups and if unequal adjusted models using the relevant covariates will be presented as well as unadjusted comparisons.

A few participants in either group may be referred to rehabilitation by their treating physician. This will be compared between groups. A number of patients may be uncontactable for final outcome measures; however, we have had a 95% rate in the current trial at 6 months postsepsis.[41] The GEE[46] will allow for missing data. If patients 'cross-over', that is, do not receive the planned intervention, they will still be analysed with the group they originally enrolled as per intention to treat.

## ETHICS AND DISSEMINATION
The trial has been registered on the Australian and New Zealand Clinical Trials Registry (ACTRN12613000528752).

Any adverse events connected to the trial will be immediately reported to one of the following three committees: RBWH Human Research Ethics Committee, The University of Queensland, Griffith University and the Australian Government Department of Health.

The results of this study will be submitted for publication to peer-reviewed ICU journals and presented at national and international ICU and/or rehabilitation conferences.

## CONCLUSION
Follow-up post-ICU stay is presently a topic of great interest with a realisation that surviving critical illness may result in poor quality of life. Patients who have had severe sepsis syndromes have been shown to have a poorer quality of life and outcome than generic ICU patients. This study will investigate whether targeted intervention from a multidisciplinary team will result in benefits and is economically feasible for this group of patients.

**Author affiliations**
[1]Burn, Trauma and Critical Care Research Centre, The University of Queensland, Brisbane, Queensland, Australia
[2]Department of Intensive Care Medicine, Royal Brisbane & Women's Hospital, Brisbane, Queensland, Australia
[3]School of Rehabilitation Sciences, Griffith University, Brisbane, Queensland, Australia
[4]Department of Physiotherapy, Royal Brisbane & Women's Hospital, Brisbane, Queensland, Australia
[5]CONROD, The University of Queensland, Brisbane, Queensland, Australia
[6]School of Medicine (Ipswich Campus), The University of Queensland, Ipswich, Australia
[7]School of Medicine, Griffith University, Brisbane, Queensland, Australia
[8]Nursing Faculty, Queensland University of Technology, Brisbane, Queensland, Australia
[9]Intensive Care Unit, Bundaberg Hospital, Bundaberg, Queensland, Australia

**Contributors** All authors participated in the manuscript preparation, writing and editing. SS participated in grant writing and protocol development.

**Funding** Funding has been provided for this study from the Royal Brisbane and Womens' Hospital Research Foundation ($40 000) and the Physiotherapy Research Fellowship, Health and Medical Research, Queensland Government ($182 000).

**Competing interests** None.

**Ethics approval** Royal Brisbane and Women's Hospital Human Research Ethics Committee (HREC; HREC/13/QRBW/17), The University of Queensland HREC (2013000543), Griffith University (RHS/08/14/HREC) and the Australian Government Department of Health (26/2013).

**Provenance and peer review** Not commissioned; externally peer reviewed.

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
