## [Reviewer comments · BMJ Open]

Some articles will have been accepted based in part or entirely on reviews undertaken for other BMJ Group journals. These will be reproduced where possible.

ARTICLE DETAILS

TITLE (PROVISIONAL)	IMPOSE: IMProving Outcomes after SEpsis: The effect of a multidisciplinary follow up service on health related quality of life in patients post sepsis syndromes, a double blinded randomized controlled trial: protocol
AUTHORS	Paratz, Jennifer; Kenardy, Justin; Mitchell, Geoffrey; Comans, Tracy; Coyer, Fiona; Thomas, Peter; Singh, Sunil; Luparia, Louise; Boots, Robert

VERSION 1 - REVIEW

REVIEWER	Elizabeth Dean University of British Columbia Canada
REVIEW RETURNED	13-Mar-2014

GENERAL COMMENTS	The work is important and addresses a major gap in our knowledge and one that is a priority given people with sepsis have better survival than they used to. The rationale and quality of the work are well supported and rigorous. This research is a priority. Re Inclusion Criteria. I would suggest a time frame since discharge for inclusion into the study, e.g., within one or two months. Re Exclusion Criteria. I would suggest exclusion of those with psychiatric or mental disabilities that would preclude being able to read, write and interpret the questionnaires. People who do not read or write English may need to be interviewed regarding questionnaire completion (this may influence reporting). In my view a strength of the trial is that there are no structured interventions; these are left to the judgment and discretion of each health professional on the team. I was pleased to see the inclusion of a nutritionist and nutritional review, and attention to mental health in the assessments. I might suggest to the team to consider other relevant lifestyle behaviors (smoking; periods of prolonged sitting and physical activity levels; sleep quality and quantity). The paper would benefit from editing throughout for typos, missing words and punctuation, abbreviation conventions, capitalization, positioning of reference numbers, etc.
---

REVIEWER	Jan Bakker Erasmus MC University Medical Center Rotterdam The Netherlands
-----------------	---

REVIEW RETURNED	07-Apr-2014
-------------

GENERAL COMMENTS	I suggest to have the power calculation provided by the authors checked by a statistician. The current number of patients that will be enrolled is rather small
--

VERSION 1 – AUTHOR RESPONSE

Reviewer 1

- 1. Inclusion Criteria I would suggest time frame since discharge for inclusion into the study**

We have added this point and included one month since discharge from hospital in the inclusion criteria – Page 8 Inclusion criteria

- 2. Exclusion criteria—I would suggest exclusion of those with psychiatric or mental disabilities. People who do not read or write English may need to be interviewed.**

Yes we would agree with this point. We actually had these as exclusion criteria but one ethics committee thought this was too biased. However they agreed eventually that this should be a basis for exclusion and so we have added this in again to the protocol. Changes on Pages 8-9

- 3. I might suggest to the team other relevant lifestyle behaviours (smoking; periods of prolonged sitting and Physical activity levels; sleep quality and quantity.**

This is a good suggestion - One of the investigators is a respiratory physician as well as an intensivist. He has a through medical system review to use in this trial and this includes question about these factors. As he is a respiratory physician he always checks about sleep quality.

Reviewer 2

- 1. I suggest to have the power calculation provided by the authors checked by a statistician**

One of our investigators (Comans) is a statistician as well as a health economist and initially planned the sample size. It was based on preliminary results from our inpatient trial on sepsis (ACTRN 1261000080844). We now have the final results for this trial and have found with only 25 in each group there was a significant effect on SF36, particularly the physical function aspect.